

# Malware detection framework based on graph variational autoencoder extracted embeddings from API-call graphs

Hakan Gunduz

Software Engineering Department, Kocaeli University, Kocaeli, Marmara, Turkey

## ABSTRACT

Malware harms the confidentiality and integrity of the information that causes material and moral damages to institutions or individuals. This study proposed a malware detection model based on API-call graphs and used Graph Variational Autoencoder (GVAE) to reduce the size of graph node features extracted from Android apk files. GVAE-reduced embeddings were fed to linear-based (SVM) and ensemble-based (LightGBM) models to finalize the malware detection process. To validate the effectiveness of the GVAE-reduced features, recursive feature elimination (RFE) and Fisher score (FS) were applied to select informative feature sets with the same sizes as GVAE-reduced embeddings. The results with RFE and FS selections revealed that LightGBM and RFE-selected 50 features achieved the highest accuracy (0.907) and F-measure (0.852) rates. When we used GVAE-reduced embeddings in the classification, there was an approximate increase of %4 in both models' accuracy rates. The same performance increase occurred in F-measure rates which directly indicated the improvement in the discrimination powers of the models. The last conducted experiment that combined the strengths of RFE selection and GVAE led to a performance increase compared to only GVAE-reduced embeddings. RFE selection achieved an accuracy rate of 0.967 in LightGBM with the help of selected 30 relevant features from the combination of all GVAE-embeddings.

Corresponding author
Hakan Gunduz,
hakan.gunduz@kocaeli.edu.tr

## INTRODUCTION

In our daily life, the use of mobile devices gradually increases as they facilitate many human needs. The increasing interest of end-users on mobile devices gives rise to service providers transferring their activities and services to mobile platforms. Due to this transformation, the number of mobile applications increases by the day. Moreover, end-users store personal multimedia data such as photos and videos, as well as confidential data such as card information, user names, and passwords on mobile devices. Attackers can generate malware on the mobile operating systems (OS) to gain financial benefits from users. Android OS is frequently targeted by malware because of both having a higher market share and more number of developed applications compared to its competitors. According to the report published by Kaspersky in 2021, the number of malware increased from

approximately 1.150M in Q1 of 2020 to approximately 1.250M in the Q2 of 2020. The rapid increase in the number of malware shows the importance of analysis methods on Android OS (https://go.kaspersky.com/rs/802-IJN-240/images/KSB_statistics_2020_en.pdf).

Malware defects the confidential information of institutions or individuals that causes material and moral damages. Therefore, intensive studies are carried out at both academic and industry levels to detect malware components effectively and continuously in Android-based devices. Literature studies on malware analysis present us with two types of analysis techniques named static and dynamic. Static analysis performs malware examinations based on source files without running them on any virtual/real devices. On the other hand, attackers can bypass the detection mechanisms of static analysis with code obfuscation. Dynamic analysis is more effective in detecting malicious activity consisting of code obfuscations. One of the cons of dynamic analysis is that malicious activity in the applications can be triggered during the conduction of analysis. In addition, running a malicious application within a sufficient time interval will have a great impact on the dynamic analysis results.

In the last decade, the analysis of Android applications was conducted automatically with the help of machine learning (ML) and deep learning (DL) models. The performances of such models were directly related to the quality of handcrafted features. To boost the predictive power and increase the generalization ability of ML models, dimensionality reduction methods reduce the size of the feature space. Feature selection is a type of dimensionality reduction method that aims to find the subset of informative features from the entire feature space. It is a process that needs human labor and domain expertise. Since DL can directly extract high-level features from instances and automatize the feature engineering process, it has gained popularity in malware detection tasks. The ability of DL models on solving complex problems is another aspect that attracts many researchers to use such models in the detection of malware.

ML and DL models perform malware detection in 3 steps:
1. The analysis of Android apk files with appropriate tools.
2. The extraction of static and dynamic features from the analyzed files.
3. The use of extracted features in model training to discriminate malware from benign

While ML models such as K-nearest neighbors, support vectors machines, naive Bayes, random forest, and decision trees (*Al-Kasassbeh et al., 2020*; *Chumachenko, 2017*; *Mahajan, Saini & Anand, 2019*) were the most preferred algorithms in malware detection, several DL models such as convolutional neural networks (CNN), artificial neural networks (ANN), and recurrent neural networks (RNN) were employed in recent detection studies (*Alzaylaee, Yerima & Sezer, 2020*; *Hemalatha et al., 2021*; *Kim et al., 2018*). Features mainly obtained from apk files of the applications such as permissions, op-code sequences, function call graphs (FCG), and Application Programming Interface (API) calls were used as inputs to these models to detect the malicious applications (*Liu et al., 2020*).

Graph-based models recently have been adopted in many tasks since they can model the latent properties between nodes and edges successfully. For instance, graph convolutional neural networks (GCNs), and graph attention networks (GANs) can extract rich representations that result in performance increases compared to the traditional ML

and DL models (*Kipf & Welling, 2016*; *Veličković et al., 2017*). The advantage of graph-based models is that they can capture the behavioral features and information accurately which lack in other methods.

Our study proposed a malware detection framework based on the Graph Variational Autoencoder (GVAE). To train and validate the framework performance, we acquired malware and benign Android applications from two public datasets. Since GVAE considered the edges between pair nodes that may contain specific information about its end-points in generating node representations, we applied it to reduce the size of API-call graph nodes extracted from Android .apk files. Thus, GVAE produced low-dimensional node representation vectors from API-call graphs to generate graph embeddings with varying sizes. We then provided GVAE-reduced embeddings to linear (SVM) and ensemble-based (LightGBM) models to realize the malware detection process and finally assessed the performances of these models with both accuracy and F-Measure metrics. To validate the effectiveness of the GVAE model on dimensionality reduction, we chose Recursive Feature Elimination (RFE) and Fisher Score (FS) as alternative selection methods. For making a fair comparison, we determined the same number of features as GVAE-reduced feature sets from node features by RFE and FS selections. In the last step, we pipelined GVAE with RFE and FS selections to create a hybrid reduction method.

The main contributions of our work can be summarized as follows:

- First of all, our malware detection framework contributes to the generation of robust node feature embeddings from the API-call graphs with the help of GVAE. GVAE handles irregularities in latent space by embedding input data to distribution rather than a point and generates the less noisy and compact node embeddings from raw node features. To the best of our knowledge, this is the first malware detection study that employs GVAE in generating node embeddings from API-call graphs.
- Our second contribution is the use of a hybrid reduction pipeline that combined GVAE with two different feature selection methods. The proposed model selected relevant feature representations from GVAE-reduced embeddings to improve the malware detection performance. The notable performance of RFE and Fisher Score in handling noisy data during the selection process is the main factor in using these methods hierarchically. To the best of our knowledge, it is the first study that combines different types of selection methods with GVAE-reduced embeddings to use in the malware detection process.
- Our last contribution is that the model used in the dimensionality reduction steps has a flexible structure. Networks (such as biological, citation, *etc.*), contain nodes with high dimensional features that can result in memory and computation issues in the training and inference phase of ML models. Our hybrid feature reduction model has a generic structure that can be applied to the aforementioned tasks to address the complexity and memory issues.

The remainder of the paper consists of four sections. "Related Work" briefly summarizes recent malware studies. "Methods" section explains the used dimensionality reduction methods, classification models, and evaluation metrics in detail. "Experimental Results"

section presents the results of all conducted experiments in order, and "Conclusion and Discussion" section concludes the paper and compares the experimental findings with recent studies.

## RELATED WORK

As mentioned before, static malware analysis is performed without running applications on virtual or real mobile devices. There are many studies in which a static analysis approach is used to detect Android malware. This approach is based on the intuition that the static attributes of applications belonging to the same malware family should be similar. The use of static analysis for the detection of malware is quite common because malware can be examined quickly without infecting any device. In addition, low analysis cost and low resource consumption can be mentioned as positive aspects of static analysis.

Many malware static analysis studies used different types of feature sets given as inputs to the ML and DL models. These features can be listed as source codes of the applications, application permissions, Application Programming Interface (API)-call graphs, and component dependency graphs (CDG). For instance; Malgenome was a pioneer study in which the static analysis approach was employed to inspect Android malware families within 1260 collected malware applications. Methods in the application source codes and application permissions in the AndroidManifest.xml were two data sources during the examination process. Malgenome study revealed that malicious applications requested SMS-related permissions such as READ_SMS, WRITE_SMS, RECEIVE_SMS, and SEND_SMS more frequently than non-malicious applications (*Zhou & Jiang, 2012*).

Drebin was another study supporting evidence in the Malgenome and used API-calls to evaluate for services in sending/receiving SMS messages (*Arp et al., 2014*). It was also one of the early studies that used machine learning models for the detection of Android malware. Permissions, activities, services, content providers, and broadcast receivers were the types of features extracted from apps in the Drebin dataset. Linear Support Vector Machines were used as an ML model to determine the families of 5560 malware applications. Drebin was been an inspiration to future malware detection studies with the specification of using different features and performing feature selection to find effective features in malware detection.

*Suarez-Tangil et al. (2014)* proposed a model named "Dendroid" that utilized control flow graphs (CFG) as model inputs. CFGs were used to extract the code structures/blocks in malware applications. The K-Nearest Neighbor model was formed with the extracted frequencies of code blocks based on each malware family. A single linkage hierarchical clustering algorithm was used to extract hierarchical similarities between malware families and the results were represented with dendrogram trees.

DroidSIFT (*Zhang et al., 2014*) was the first study that employed a graph-based method for malware family classification. This study extracted methods and API-calls from the source code of applications to express apps with weighted contextual API-based graphs. The classification process considered API-calls that match the permissions requested from the user as well as security-related API-calls. To perform the classification process, the

similarity value between each graph obtained from a new app and the graphs of different malware apps was computed.

Image processing-based features were also employed in static analysis. In *Iadarola et al. (2020)*, malware applications were converted to grayscale and binary images before passing them through four different image filters. An accuracy rate of 96.9% has been demonstrated with the combination of the feature representations obtained from filters and the Random Forest model.

Deep learning architectures were also used together with the static analysis approach in recent malware detection studies. *Sewak, Sahay & Rathore (2018)* blended different types of deep learning architectures and extracted feature representations from deep layers automatically. Their model achieved accuracy and false-positive rates of 99.21% and 0.19% respectively. Another study presented a shallow malware detection model to handle the overfitting of DL models. This model used the combination of a Convolutional Neural Network (CNN) with given a sequence of op-code instructions as input data. The proposed model achieved a 95% rate of accuracy over a dataset that contained nearly 70,000 instances. In *Kang et al. (2020)*, two different dataset samples were derived from the .dex files of applications using image processing techniques. The first dataset created feature sets from the entire .dex files, while the second dataset created features considering only the data part of the files. The CNN model was trained with the extracted datasets and malware families were predicted with an accuracy rate of 91%.

Autoencoder is the other deep learning architecture actively studied in cyber-security domain for anomaly detection (*Xu et al., 2021*), data generation (*Kabore et al., 2021*), and dimensionality reduction (*Haseeb et al., 2022*). For example, several autoencoder models have been utilized in intelligent Network Intrusion Detection Systems (NIDS) to handle zero-day attacks with high accuracy (*Song, Hyun & Cheong, 2021*). The variational autoencoder (VAE) has been used to generate intrusion data in a generative manner to cover the imbalanced data problem generally seen in many intrusion detection systems (*Lopez-Martin, Carro & Sanchez-Esguevillas, 2019*; *Vaiyapuri & Binbusayyis, 2020*). *Yousefi-Azar et al. (2017)* employed autoencoders to generate code vectors that captured latent representations of different feature sets. Trained autoencoders generated distinguished features from original features in an unsupervised fashion for malware classification and decreased the computational complexity of the proposed model significantly.

Recently, graph neural networks (GNN) have gained popularity in the cyber-security domain, especially in malware detection tasks. For instance, *Xu, Eckert & Zarras (2021)* proposed a GNN-based malware family classification model that transformed function call graphs into dense embedding vectors to maintain the relationships between functions in the applications. The accuracy rates of the models increased up to 99.6% in malware detection and 98.7% in malware classification tasks. *Gao, Cheng & Zhang (2021)*'s study presented a model named "Gdriod" for malware classification. This study made use of a GNN model on a heterogeneous graph to model edge-based relationships between applications mapped APIs. The success rate of the proposed model measured 98.99% in terms of accuracy in the malware detection task. Our recent study on malware detection extracted API-call graphs from Android apk files and detected malicious applications over

Android-based devices placed at intelligent transportation systems. Our work constructed two types of node features as Node2Vec embedding and network properties for each node in API-call graphs. Graph attention networks (GAN) were trained with extracted feature sets and the combination of GAN and Node2Vec features showed the best performances over the entire feature set and GNN combinations (*Catal, Gunduz & Ozcan, 2021*).

## METHODS

This section provides information on the proposed framework and explains the details of each component of such framework. The proposed malware detection framework is designed as an end-to-end model that takes the Android .apk files as model input and classifies these files as benign or malware in the output. Figure 1 presents the graphical representation of the framework. The proposed framework consists of four sequential steps. In the first step, training and test apk files were collected from two public datasets. The details about the used datasets were explained in the next subsection. In the second step, API-call graphs, which represented caller-callee relationships between the methods in a source code, were created from .apk files with the help of the Androguard tool. After the call graph generation, the Node2Vec model produced 100-dimensional features for each node in the graphs. The third step of the framework realized the dimensionality reduction process. Graph Variational Autoencoder (GVAE) was first applied to obtain reduced node embeddings with varying sizes. Following the production of node embeddings, the graph embedding vector was constructed by averaging all the node embeddings in the graph. At last, recursive feature elimination (RFE) and Fisher score (FS) were used to reduce the dimensions of the graph embedding vector. As stated in the introduction, the RFE and FS methods were used to demonstrate the effectiveness of the GVAE method in size reduction. RFE was determined as an alternative model to GVAE due to its success in handling the dependencies and collinearity between the attributes that are also present in the graph-structured data. On the other hand, FS was chosen as another comparison method since it considered both positive and negative class samples during the computation of feature relevances and directly assessed the relevance between each feature and class labels. The last step of the framework covered the model training phase. As stated in *Liu et al. (2020)* and *Pan et al. (2020)*, the vast majority of malware detection studies employed support vector machines (SVM) and ensemble learning models in the classification process. Our framework utilized SVM and LightGBM models to assess the effectiveness of the reduced graph embeddings. The predictive performance of these models was compared in terms of accuracy and F-Measure metrics.

The details of each component in the framework were explained in the subsections below.

### Dataset

To assess the performance of our proposed framework, we used two open-access datasets from Canadian Institute for Cybersecurity website (https://www.unb.ca/cic/datasets, accessed on 10 December 2021). The first dataset is ISCX-AndroidBot-2015 which comprises 14 botnet families with 1929 instances (https://www.unb.ca/cic/datasets/android-botnet.html, accessed on 10 December 2021). Since this dataset does not include any

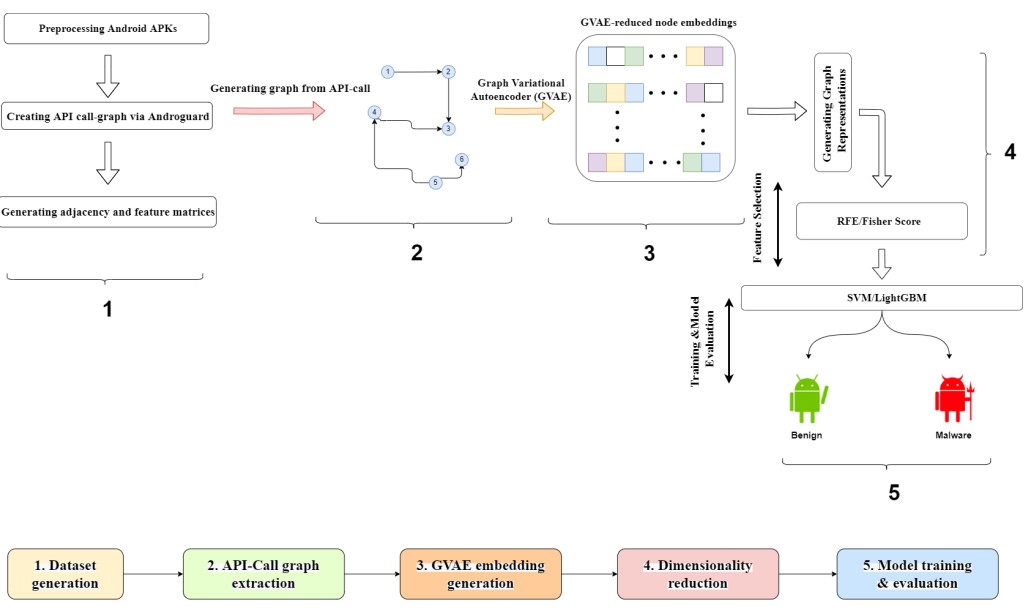

**Figure 1** Graphical representation of the proposed framework.

**Table 1** Information about datasets.

| Dataset | #instances | Type |
| --- | --- | --- |
| CICMalDroid | 1929 | benign |
| ISCX-AndroidBot-2015 | 1843 | malware |

benign instances, we provide benign apps from another dataset named CICMalDroid (https://www.unb.ca/cic/datasets/maldroid-2020.html, accessed on 10 December 2021). We acquired 1,795 benign instances from CICMalDroid and finally devised a dataset consisting of 3,724 instances (Table 1).

After generating the dataset, we extracted the API-call graphs from application .apk files. We used the Androguard tool for the API-call extraction process. We first determined the sensitive Android APIs from all API sets. Selected sensitive APIs are "Landroid.accounts", "Landroid.app", "Landroid.bluetooth", "Landroid.content", "Landroid.location", "Landroid.net", "Landroid.nfc", "Landroid.provider", "Landroid.telecom", and "Landroid.telephony". We then created the nodes of call graphs by representing caller-callee relationships between the methods of the sensitive APIs. Following the creation of API-call graphs, we used the Node2Vec for generating 100-dimensional features for each node in the graphs.

## Graph Variational Autoencoder (GVAE)

An autoencoder is a neural network architecture that reconstructs the samples given to the input layer at the output layer. Variational autoEncoder (VAE), on the other hand, is a generative autoencoder model that forces the distribution of samples in the hidden space to a normal distribution (*An & Cho, 2015*). VAE composes of two separate components as

encoder and decoder. The encoder creates a hidden representation vector **h** from the input vector **x** in the hidden space, while the decode makes use of **h** vector to reconstruct the **r** output with the decoder network. VAE expresses the vector **x** in the input layer in terms of 2 parameters in the hidden space. These parameters are the mean and standard deviation (sd), which are the descriptive statistics of the learnt normal distribution. VAE generates new low-dimensional samples with the learnt mean and sd vectors after model training. Although mean and sd values are deterministic, samples generated from these values are probabilistic.

A graph variational autoencoder (GVAE) is proposed by *Kipf & Welling (2016)* for representation learning that operates the VAE over the graph data. GVAE basically generates new graphs from original input graphs. Due to having irregularities in graph-structured data, VAE is not directly applied to form feature representations for each node in the graphs. GVAE uses adjacency and feature matrices in generating node embeddings. While adjacency matrix $A$ represents the neighborhood relationships between each node, feature matrix $X$ extracts the feature information of each node from the input graph. The encoder component of GVAE consists of two consecutive GCN layers to generate the latent variable $Z$ as output. The first GCN layer takes $A$ and $X$ matrices as inputs and creates $\sim A$ and $\sim X$ matrices to be inputted to the second GCN layer. The output of the second GCN is $\mu$ and $log\sigma$ vectors. Low dimensional Z matrix is calculated with generated $\mu$ and $log\sigma$ matrices using re-parameterization trick. The decoder component of GVAE is defined by an inner product between latent variable $Z$ and the output of decoder component is a reconstructed adjacency matrix.

## Recursive feature elimination

Recursive feature elimination (RFE), which is a wrapper feature selection method, uses ML methods such as SVM and GBM to assign the feature relevance scores (*Granitto et al., 2006*). RFE initially builds a model from whole features and calculates a feature importance score for each feature. After that, the feature with the least importance score is removed from the feature space and the model is reconstructed with the remaining features for computing new importance scores. This procedure is maintained up to the predefined number of features retains in the dataset. Hence, the desired feature count is a hyper-parameter for RFE. Another parameter to be specified in the RFE is the ML model, which is employed in calculating the feature importance scores. SVM is a favored algorithm for RFE to its high accuracy and robust generalization ability. At each iteration of the RFE, the Linear SVM model is trained to assign a weight coefficient to each feature. Since the feature with the lowest weight could have the least effect on the classification, this feature can be ignored in the next iteration. In the case of high dimensional feature space, more than one feature may be omitted per each iteration of RFE (*Gunduz, 2021a*).

## Fisher score

The Fisher score is a filter-based method that aims to measure the relevance between each feature and the class label to select the informative features. The Fisher score utilizes the mean and standard deviation values of the features for each class in computing feature

relevances. The equation of the Fisher score is shown below:

$$f(k) = \frac{\sum_{j=1}^{C} n_j (\mu_j^k - \mu^k)^2}{\sum_{j=1}^{C} n_j (\sigma_j^k)^2}. \tag{1}$$

In Eq. (1), $\mu_j^k$ denotes to the mean of the k-th feature in the j-th class while $\sigma_j^k$ indicates the variance of the k-th feature in the j-th class. $n_j$ refers to the total instance counts in the j-th class. $\mu^k$ shows the mean of the k-th feature. In the Fisher score selection, the Fisher scores of all the features are sorted in descending order, and the desired number of features are selected starting with the high-scoring features (*Sun et al., 2021*; *Gunduz, 2021b*).

## Support Vector Machines (SVM)

Support vector machines (SVM) is a machine learning algorithm used to solve both classification and regression problems. SVM aims to find an optimal hyperplane that separates the instances of two classes in a binary classification task. SVM creates this hyperplane using different kernel functions in datasets in which the number of dimensions is more than the number of instances. However, some problems consist of data points that cannot be separated linearly. Therefore, SVM projects the patterns in non-separable data into a new space and looks for a hyperplane in the new space. SVM uses the kernel functions to project the linearly non-separable dataset to larger dimensional spaces that can be linearly separated. The linear separation of the dataset is realized with a certain error because of the noisy and complex structure of the data. In the case of linear separation with a certain error, a slack-bound approach is used to separate the two-class dataset. To reduce the probability of misclassification, the problem turns into an optimization problem by performing transformations in the linear separation case with the help of the C coefficient. Lower C values can cause under-fitted models that may have more misclassified samples, while higher C values tend to rise the variance of the model and lead to overfitting (*Huda et al., 2016*).

## LightGBM

Boosting is an ensemble model that constructs a single strong learner from a predefined number of base learners. Boosting trains a group of learners with the same dataset instances, but adjusts the weights of the instances according to the errors of the final prediction. The intuition behind the boosting is to empower models to focus on instances that are hard to predict.

LightGBM is a fast, distributed, and high-performance boosting model built on decision trees. It is a typical gradient boosting strategy that utilizes many weak decision trees. Opposite to the bagging strategy, LightGBM iteratively combines models. Boosting models have two formation approaches, level-oriented and leaf-oriented, during the iterative training of each decision tree. The level-oriented approach maintains the balance property during tree expansion, whereas the leaf-oriented approach continues to split the biggest loss-decreasing leaf. LightGBM makes use of a leaf-oriented approach that considers both losses in a particular tree split and the contribution of this splitting to the entire loss. Therefore, it forms the trees with lower errors rather than a level-oriented tree growing (*Ke*

*et al., 2017*). The training time of a simple decision tree is directly related to the number of possible node splits. Small variations in splitting often do not make a big distinction in model performance. LightGBM utilizes this case by grouping the features into several bins and splitting them into the bins instead of the features. This property can decrease the computational complexity and result in reductions in model training time.

The basic parameters to be determined in the LightGBM are the number of learners, learning rate, and max-depth. The number of learners is the number of iterations used in setting up the ensemble model. A high number of iterations can lead to overfitting while a low number prevents us from learning patterns. Learning rate (lr) is a value between 0 and 1 for scaling generated trees. A smaller lr can help better predictive power. However, it can increase model training time and result in possible overfitting. Max-dept is used to limit the depth of the tree to be built. It should be optimized to avoid overfitting. Too much branching will cause overfitting, and too little branching will cause underfitting.

### Evaluation metrics

Although accuracy is a common measure in performance evaluation, it has a lack of ability in the assessment of class discrimination. F-measure is an alternative metric to use in the validation of the class-based model performance. The computation of the accuracy and F-Measure is directly related to Confusion Matrix (CM) in which basically presents the number of correct and incorrect predicted instances per class (Table 2). True positive (*tp*), false positive (*fp*), false negative (*fn*), and true negative (*tn*) are the values used to compute aforementioned metrics (*Gunduz, 2021b*).

Accuracy is defined as the ratio of the number of accurate predictions to the total number of instances. However, when the ratio between the *fp* and *fn* becomes very large, F-measure needs to handle the job in performance evaluation.

F-measure considers precision and recall by taking the harmonic mean of both metrics. Therefore, false positive and false negative samples are involved in the assessment of class discrimination. Based on the confusion matrix, F-measure is computed as follows:

$$\text{precision} = \frac{tp}{tp + fp} \tag{2}$$

$$\text{recall} = \frac{tp}{tp + fn} \tag{3}$$

$$\text{F-Measure} = \frac{2 \times precision \times recall}{precision + recall}. \tag{4}$$

## EXPERIMENTAL RESULTS

As aforementioned before, experiments were conducted with a dataset formed from the combination of two public datasets. Seeing that graph-structured data needed a high computational resource, experiments were realized on a PC with GTX 1070 graphics

**Table 2  Confusion matrix for two-class classification.**

| Actual/Predicted as | Positive | Negative |
|---|---|---|
| Positive | *tp* | *fn* |
| Negative | *fp* | *tn* |

**Table 3  Parameters of proposed GVAE model.**

| Parameter | Value |
|---|---|
| #epoch | 100 |
| #hidden units in GCN | {20, 30, 40, 50} |
| #GCN layers | 2 |
| Dropout rate | 0.2 |
| L2-regularization rate | 0.01 |
| Optimizer | Adam |

processing unit (GPU) support. Pytorch-Geometric framework (*Fey & Lenssen, 2019*) of the Pytorch was employed to build graph variational autoencoder (GVAE) models. Compared to Keras and Tensorflow, Pytorch presents great and diverse opportunities for building graph neural networks with the help of the Pytorch-Geometric package.

In the first experiments, the GVAE model reduced the size of the node features and generated low-dimensional node embeddings by considering the adjacency relations between the nodes as well as neighbors node features. The proposed GVAE architecture used the GCN network in its encoder component. Table 3 presented the hyperparameters of the generated network architecture. The training of the GVAE model was realized in 100 epochs with 64 batch sizes. While there were diverse options for the selection of optimizer functions including Adagrad, AdaBelief, and Rmsprop, Adam was chosen as the optimizer due to its ability to achieve convergence quickly with high accuracy (*Bock & Weiß, 2019*). Both dropout and regularizer layers were also attached subsequently to GCN layers for avoiding overfitting during model generation.

GVAE reduced the sizes of node features from 100 to 20, 30, 40, and 50, respectively. After producing low-dimensional node embeddings, each graph was represented with a graph embedding vector by averaging the node embedding vectors. SVM and LightGBM models were trained with graph embedding vectors, and the performances of the models were evaluated with the 10-fold cross-validation (CV) method. Despite the most preferred approach in performance assessment being a hold-out method, this method cannot consider all instances in the dataset and can cause biases in performance evaluation. In addition, cross-validation is simple to comprehend and is less susceptible to biased prediction in the evaluation of the model success. A grid search was conducted on the parameters specified in Tables 4 and 5 in company with the CV to find out the optimal parameter set. To assess the statistical properties of the obtained results, the Wilcoxon signed rank test was utilized with a 0.05 significance level.

The results in Table 6 showed that LightGBM had more successful performance than SVM in terms of accuracy and F-Measure metrics. LightGBM reached 0.943 accuracy with

**Table 4  Parameter space of LightGBM.**

| Parameters | Value |
|---|---|
| number of learners | {100,200,500,1000} |
| learning rate | {0.1,0.01} |
| L2-regularizer | {0.001,0.0001} |
| max_depth | {7,9,11} |

**Table 5  Parameter space of SVM.**

| Parameters | Value |
|---|---|
| Kernel Type | {rbf,poly} |
| Regularization (C) | {0.5,0.1,1,2,4,8} |

**Table 6  Classification results with GVAE-reduced node embeddings.**

| #Features | LightGBM | | SVM | |
|---|---|---|---|---|
| | Accuracy | F-Measure | Accuracy | F-Measure |
| 20 | 0.917 | 0.875 | 0.909 | 0.865 |
| 30 | 0.943 | 0.909 | 0.927 | 0.892 |
| 40 | **0.943** | **0.912** | **0.935** | **0.902** |
| 50 | 0.937 | 0.901 | 0.927 | 0.889 |

0.912 F-Measure rates using only 40 features. SVM also showed satisfactory performance with 40 features that resulted in an accuracy rate of 0,935 with an F-measure rate of 0.912.

After the classification process with GVAE-reduced embeddings, the second experiments used raw node features directly in the classification process. Since each node in the graph included 100 features, each graph was represented by a 100-dimensional vector by averaging such node features. Following the formation of the graph vectors, feature subsets with varying sizes were created with RFE and FS selections. RFE benefited from SVM and LightGBM models for the computation of feature-relevance scores. To make a fair comparison with the results obtained in the first experiments, the size of graph vectors was reduced from 100 to 20, 30, 40, and 50, respectively. SVM and LightGBM models were trained with the obtained selected relevant features and their classification performances were evaluated with 10-Fold CV.

Table 7 showed that the highest classification accuracy was achieved with 50 features selected by RFE. With this feature set, LightGBM and SVM had accuracy rates of 0.907 and 0.886, respectively. The combination of FS with LightGBM underperformed slightly than RFE selection with an accuracy of 0.895 (Table 8). The performance of SVM stayed behind LightGBM and obtained 0.874 accuracy with a subset of 50 FS-selected features.

In the last experiment, two-dimensionality reduction methods used in the previous experiments were blended. In order to achieve this, all GVAE-reduced embedding sets (20,30,40, and 50) were concatenated. The combination of all embedding features resulted in a 140-dimensional vector for each graph. After the expansion of feature space with

**Table 7  Classification results with raw node features (RFE-selected).**

| #Features | LightGBM | | SVM | |
|---|---|---|---|---|
| | Accuracy | F-Measure | Accuracy | F-Measure |
| 20 | 0.873 | 0.802 | 0.851 | 0.779 |
| 30 | 0.892 | 0.831 | 0.870 | 0.808 |
| 40 | 0.901 | 0.846 | 0.883 | 0.823 |
| 50 | **0.907** | **0.852** | **0.886** | **0.835** |

**Table 8  Classification results with raw node features (FS-selected).**

| #Features | LightGBM | | SVM | |
|---|---|---|---|---|
| | Accuracy | F-Measure | Accuracy | F-Measure |
| 20 | 0.861 | 0.791 | 0.841 | 0.767 |
| 30 | 0.883 | 0.820 | 0.862 | 0.798 |
| 40 | 0.889 | 0.834 | 0.872 | 0.811 |
| 50 | **0.895** | **0.841** | **0.874** | **0.825** |

**Table 9  Classification results with the combination of GVAE-reduced embedding and RFE selection.**

| Model | #Features | Accuracy | F-Measure |
|---|---|---|---|
| **LightGBM** | **30** | **0.967** | **0.934** |
| **SVM** | **40** | 0.955 | 0.924 |

all embedding sets, the dimensions of feature vectors were reduced *via* RFE selection. The main reason for using RFE is that it outperformed FS in the previous experiments. Classification results obtained with the combination of the GVAE-reduced embeddings and RFE selection were shown in Table 9.

The results obtained with the combination of GVAE-reduced embeddings were higher than those obtained with the individual GVAE-reduced features sets. When RFE selection was made on the combined GVAE-reduced embeddings, the most successful classification result was again obtained with LightGBM. Classification accuracy was up to 0.967 with LightGBM, while the accuracy rate was realized in 0.955 with SVM. LightGBM achieved this result with 40 informative features. On the other hand, SVM reached the highest success rate with selected 30 features.

## CONCLUSION AND DISCUSSION

Even though numerous studies have been realized on malware detection using ML and DL models, detecting malware effectively using graph variational autoencoders remains an unexplored topic area in the cyber-security domain. Our study used API-call graphs for malware detection and performed different dimensionality reduction methods on node features to find malicious code patterns. The first experiments utilized GVAE to extract low-dimensional node embeddings of several sizes from API-call graphs. The next experiments applied RFE and FS selections to select informative feature sets with the

same sizes as GVAE-reduced embeddings. The results with RFE and FS selections revealed that LightGBM achieved the highest accuracy (0.907) and F-measure (0.852) rates using 50 features. SVM again showed sufficient performance with an accuracy rate of 0.886. When both models used GVAE embeddings as model inputs, there was an approximate increase of 4 percent in their accuracy rates. Same performance increases could also be seen in F-Measure rates that directly indicated the improvement in the discrimination power of the models. LightGBM and SVM reached the best accuracy rates with 40 and 30 reduced features, respectively. The last conducted experiment combined the strengths of RFE selection and GVAE that led to the performance rise compared to only GVAE-reduced embeddings. RFE selected 30 relevant features from the combination of all GVAE-reduced features and boosted prediction accuracy to 0.967 in LightGBM. SVM also reached an accuracy of 0.955 with 0.921 F-Measure scores with 40 features that resulted in a nearly %2 increase on both performance metrics compared to only GVAE-reduced features.

All conducted experiments revealed that the proposed hybrid size-reduction framework had two prominent properties that helped to achieve the best results compared to all individual models. The first property is that GVAE uses the GCN model in its encoder component. GCN considers adjacent nodes as well as node features during the generation of node embeddings. The second property of the framework is that RFE employs LightGBM in computing feature importance scores. LightGBM is an efficient model for reducing variance and preventing overfitting during the computation of feature relevances. Obtained test results confirmed that the proposed framework can effectively detect malware with high accuracy and F-Measure scores.

The experimental results we obtained were also compared with the results of the recent malware studies that had deployed DL and ML models in the detection process. Recent survey articles presented the dominance of static analysis in the detection/classification process due to the ease of finding malware code structures without running on real devices. Moreover, most of these studies employed the features extracted from source code files such as Android permissions, Op-code sequences, API-call sequences, and API-call graphs in malware detections. Classification performances of recent studies are presented in Table 10. When the results were examined, it was concluded that the performances of the proposed models in these studies varied between 0.90 and 0.99 in terms of accuracy and F-Measure rates. In addition, these studies benefited from DL models in feature extraction and classification steps.

Considering the studies using the same feature set as in our study, it was seen that the highest success rate was achieved *Pektaş & Acarman (2020)* that compared the performances of different graph embeddings on CNN models. Since this study trains a shallow CNN model with a relatively small number of instances (5560 samples), it is not feasible to build such a model due to the chance of increasing overfitting. Another difficulty faced in this study is that the CNN model has many trainable parameters and the determination of the best parameter setting is a time-consuming process. Unlike the aforementioned study, our study used a deep learning model, GVAE, during the extraction of low dimensional embeddings. After dimensionality reduction with GVAE, each application was represented by vectors with a maximum of 50 dimensions. Reducing the feature space of the data has

**Table 10  Classification results of recent malware studies.**

| Models | Feature Set | Accuracy | F-Measure |
|---|---|---|---|
| SVM, kNN (Zhao et al., 2015) | Permissions, API-calls | 0.975 | NA |
| RF, SVM (Canfora et al., 2015) | n-opcode of classes.dex | 0.965 | NA |
| CNN (Ganesh et al., 2017) | Permissions | 0.930 | NA |
| MKL (Narayanan et al., 2018) | APIs' permissions | NA | 0.985 |
| DBN (Li et al., 2018) | API-calls | 0.900 | NA |
| CNN (Amin et al., 2019) | Risky permissions | 0.974 | 0.974 |
| BiLSTM (Ma et al., 2020) | API-call sequences | 0.972 | 0.982 |
| GE+CNN (Pektaş & Acarman, 2020) | API-call graphs | 0.988 | 0.986 |
| LightGBM (Al Sarah et al., 2021) | APIs' permissions | **0.990** | **0.980** |
| GAT (Catal, Gunduz & Ozcan, 2021) | API-call graphs | 0.961 | 0.948 |
| GVAE+LightGBM (Proposed study) | API-call graphs | **0.967** | **0.934** |

also enabled the optimum parameters of the models to be found in a short time. Our study also benefited from the LightGBM model, which reduces overfitting by adjusting the model variance at each step during training. Our previous study (Catal, Gunduz & Ozcan, 2021) trained the graph attention network model with the dataset used in this study and reached an accuracy rate of 0.96 using 100-dimensional node features. Although the classification performance of the previous study was close to this study, our proposed model achieved this performance with only 30 features.

Experimental studies have some limitations and threats to validity. In this study, experimental setups were trained with two open-source datasets. The performance of the proposed model on other datasets might be slightly different; however, we do not expect too many variations in the performance. Different researchers also might develop new malware detection frameworks models using novel deep learning architectures and achieve better performance results than the one reported in this study.

This study focused to enhance the performance of malware detection models using a novel dimensionality reduction method. The proposed framework fused the GVAE and RFE. Experimental results presented that the execution of RFE selection on GVAE embeddings provided remarkable results. In addition, the proposed framework has a generic form that can be widened to diverse domains including graph-structured data types. Tasks in bioinformatics and recommendation systems are some examples of these domains where our framework can be adopted. Future work will conduct research on the utilization of the DL and ML models in malware detection systems from the point of view of Explainable Artificial Intelligence (XAI).

## Funding
The authors received no funding for this work.

## Competing Interests

The authors declare there are no competing interests.

## Author Contributions

- Hakan Gunduz conceived and designed the experiments, performed the experiments, analyzed the data, performed the computation work, prepared figures and/or tables, authored or reviewed drafts of the paper, and approved the final draft.

## Data Availability

The code is available at GitHub:

https://github.com/hakangunduz86/Malware-Detection-with-GVAE.

The data is available at the Canadian Institute for Cybersecurity: https://www.unb.ca/cic/datasets/.

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
