# Peer review of "Malware detection framework based on graph variational autoencoder extracted embeddings from API-call graphs"

_PeerJ Computer Science, doi:10.7717/peerj-cs.988_

## Round 0.1 · original submission · Minor Revisions

We have completed the review process of the paper and reviewers have suggested Minor Revisions. You may resubmit the revised manuscript after carefully addressing all the concerns raised by the reviewers.

Reviewer 1 has requested that you cite specific references. You may add them if you believe they are especially relevant. However, I do not expect you to include these citations, and if you do not include them, this will not influence my decision.

Reviewer 1 ·

Basic reporting

Thias work presents and interesting research on applying GVAE to dimensionality reduction, followed by some classic ML models applied to the resulting features. Some comments:
- Why to choose linear SVM and LightGBM and not others.
- Highlight all assumptions and limitations of your work.
- Conclusions should provide some lessons learnt.
- Related works section does not mention recent research effors in feature characterization in NIDS using also clustering and contrastive learning, as well as, review works on VAEs. Authors are advised to refer to the following related articles to add some discussions: [1] Variational data generative model for intrusion detection, Knowledge and Information Systems, 2018 [2] Analysis of Autoencoders for Network Intrusion Detection, Sensors, 2021. [3] Network Intrusion Detection Based on Supervised Adversarial Variational Auto-Encoder With Regularization, IEEE Access, 2020 [4] Supervised contrastive learning over prototype-label embeddings for network intrusion detection, Information Fusion, 2022

Experimental design

The experimental design and the descriptions of experimental results is well done and sufficient.

Validity of the findings

Accuracy anf F1 has been obtained, and to assess the significance of results a Wilcoxon Signed Rank Test was applied using a 0.05 significance level. The results are correctly analyzed and are significant. To obtain some additional metrics eg recall, precision, etc,...would be fine but it is not strictly necessary. Table 9 of classification results of recent malware studies is interesting to position the results in perspective with others.

Reviewer 2 ·

Basic reporting

This study proposes a malware detection model based on API-call graphs and uses Graph Variational Autoencoder (GVAE), which is a variant of graph neural networks, to reduce the size of graph nodes extracted from Android apk files.
Basic reporting is good. However, literature work should be added more from the year 2020-2021

Experimental design

Authors did not mentioned why they use Recursive Feature Elimination (RFE) and Fisher Score (FS) to conduct feature selection in comparison to other studies. Please explain why you used these methods. Furthermore, Is the analysis is sufficient enough by using SVM as a model., why not considering other well-known classifiers that classifies better than SVM?

Please explain list of feature selection in tabular format.
In table 6 it seems that increasing number of features will increase accuracy, which is normally not true. So, what will be the accuracy at 60 features?

Validity of the findings

Please explain the parameter space of LightGBM in detail.

Reviewer 3 ·

Basic reporting

The paper is well written and the content is well connected. In terms of basic reporting I would recommend following corrections:
1) The abstract of the paper is lengthy. Please reduce the content of the abstract and omitting detailed methodology and only report the primary results.
2) The Figure methodology diagram should be re-drawn as a block diagram. The diagram should be divided in to different connected components.
3) The contributions of the paper should be written in bullet points or with numbering.
4) The paragraph after the contributions should be re-written.
5) The API extraction from the data set is not explained. How did you extracted the API calls? Which tools did you used? This should be explained.

Experimental design

The experiments are presented clearly, however, i have concern with the explanation of dataset. Please address the following:
1) Include a section which explains the dataset. I do not see the number of benign apps being used. Is the dataset balanced or not?
2) Include a table to present/explain the dataset.

Validity of the findings

Satisfied.

---

## Round 0.2 · accepted · Accept

Congratulations, based on the reviewer's recommendation your paper has been accepted.

Reviewer 1 ·

Basic reporting

It is ready for publication

Experimental design

It is ready for publication

Validity of the findings

It is ready for publication

Additional comments

It is ready for publication

Reviewer 2 ·

Basic reporting

Satisfied

Experimental design

Changes are made as mentioned

Validity of the findings

Satisfied

Additional comments

Suggested changes are modified.